# Muscular Fatigue and Quadriceps-to-Hamstring Ratio in Alpine Skiing in Women over 40 Years

**DOI:** 10.3390/ijerph20085486

**Published:** 2023-04-12

**Authors:** Aljoscha Hermann, Vera Christl, Valentin Hastreiter, Patrick Carqueville, Lynn Ellenberger, Veit Senner

**Affiliations:** 1Professorship of Sport Equipment and Materials, TUM School of Engineering and Design, Technical University of Munich, 85748 Garching, Germany; vera.christl@tum.de (V.C.); valentin.hastreiter@tum.de (V.H.); senner@tum.de (V.S.); 2Institute of Sports Science, Faculty of Economics and Social Sciences, Eberhard Karls Universität Tübingen, 72074 Tübingen, Germany; 3Swiss Council for Accident Prevention BFU, Hodlerstrasse 5A, 3001 Bern, Switzerland; l.ellenberger@bfu.ch

**Keywords:** muscle activity muscle fatigue, injury, alpine skiing, EMG, textile sensors, safety, quadriceps-to-hamstring ratio

## Abstract

(1) Background: In alpine skiing, senior athletes and especially women have a high risk of knee injury. This may also be related to muscular fatigue (MF) of the knee-stabilizing thigh muscles. This study investigates both the evolution of muscle activity (MA) and of MF of the thighs throughout an entire skiing day. (2) Methods: *n* = 38 female recreational skiers over 40 years of age performed four specific skiing tasks (plough turns, V-steps uphill, turns with short, and middle radii) at specific times, while freely skiing the rest of the day. Surface EMG of the thigh muscle groups (quadriceps and hamstrings) was measured using special wearables (EMG pants). Apart from standard muscle activity parameters, the EMG data were also processed in the frequency domain to calculate the mean frequency and its shift over the day as a metric of muscle fatigue. (3) Results: The EMG pants showed reliable signal quality over the entire day, with BMI not impacting this. MF increased during skiing before and for both muscle groups significantly (*p* < 0.006) during lunch. MF, however, was not reflected in the quadriceps–hamstrings ratio. The plough manoeuvre seems to require significantly (*p* < 0.003) more muscle dynamics than the three other tasks. (4) Conclusion: MF may be quantified over an entire skiing day and thus fatigue information could be given to the skier. This is of major importance for skiers at the beginner level dominantly performing plough turns. Crucial for all skiers: There is no regenerative effect of a 45-min lunch break.

## 1. Introduction

Recreational alpine skiing is a popular sport performed by individuals of all levels of experience, fitness, and age. In skiing, more than four of five injuries are non-contact injuries [1] and, as such, could be preventable by technical measures, training, and consciousness of the skier about the physical and mental requirements of skiing. All interventions to reduce the injury risk need an extensive understanding of the manifold factors which influence this risk. Besides external influences, such as weather conditions [2], trail conditions [3], and equipment [4,5,6], internal factors of the skiing individual are important influences. Some of these factors can be altered by the skier, such as the physical condition [7,8], skiing skill level [9] and alcohol use [10]; others, such as age [9,11] and sex [12,13], cannot be influenced. With respect to the injury location, the knee is the most frequently injured body part in skiing [14] and females are reported to have a higher knee injury prevalence than males [14,15].

This study focuses on muscle activity of the thigh muscles, more specifically the quadriceps and hamstring muscles and their fatigue behaviour over a skiing day. Thigh muscles are essential for skiing and, if well trained, can stabilise and protect the knee in critical loading situations [16]. In sports, it has been shown that the presence of fatigue has a gender-specific effect, modifies biomechanics [17,18], and reduces protective musculoskeletal mechanisms [19]. Fatigue of thigh muscles can lead to larger anterior tibial translations [20] and thus to an increased risk of anterior cruciate ligament (ACL) injury (even more for females) [21]. Decision-making in unanticipated conditions (e.g., catching the ski in the snow) in a fatigued condition is also considered to increase ACL injury risk [22]. The effect of fatigue specifically on injury risk in skiing, however, is still unclear. Several authors report different relations between fatigue and injuries, either finding no relation between those factors or that fatigue is a risk factor for injuries [23,24,25,26]. Many of these studies show limitations in the methodological approach, e.g., the use of retrospective subjective questionnaires to ask the injured skiers in the hospital about their perceived fatigue within several days after the incident [24,25]. A limitation of such questionnaires is that it may be possible that skiers do not perceive fatigue (at least upcoming fatigue) due to an excited state. It is also unclear how fatigue evolves throughout the skiing day and if regenerative breaks, such as the lunch break, have a positive effect. This knowledge could help to create awareness of such facts for example in ski school lessons or by means of wearable feedback systems. A wearable system providing an objective metric of fatigue would also allow investigation of the relationship between fatigue and injury risk without the aforementioned limitation.

Surface electromyography (EMG) is commonly used to measure muscle activity. Several mathematical methods have been developed to quantify the level of muscle fatigue from myoelectric signal patterns [27]. Processing the EMG signals in the frequency domain using fast Fourier transformation is considered to be the gold standard for assessing muscle fatigue from surface EMG signals. Muscle fatigue manifests itself in a downward shift of the mean or median frequency of the EMG signal frequency spectrum [28,29]. Moreover, the EMG signal amplitude shows an increase as a result of muscle fatigue [27,30].

As longer termed several hours EMG measurements with traditional wired surface electrodes may result in some discomfort for the subjects, the use of wearable EMG pants with integrated textile electrodes allows more comfortable recording of thigh muscle activity with a minimum of effort. Several studies [31,32] have shown such pants to be a practical and flexible alternative to traditional surface electrode-based methods in sports environments. In recent years, the price level of such pants has become affordable and the handling requires no prior measurement experience. With this in mind, the idea is obvious to collect surface EMG information to provide feedback to the skier on his or her fatigue level, information on the eventually unbalanced performance of the left and right leg, and on the quality of a quadriceps–hamstring co-contraction as an indicator of knee stabilisation. From the point of view of our currently ongoing developments to improve the safety function of ski bindings, the application of such kind of wearable would allow using muscle activity as one of the risk predictors (among several others such as skiing velocity or the actual knee flexion angle) in order to electronically adjust the ski binding’s release level (so-called mechatronic ski binding design [31]).

This study aimed to investigate how muscular fatigue of the quadriceps and hamstring muscles evolves throughout the skiing day in female skiers. Moreover, we wanted to test if EMG pants are a valid measurement tool for women with different body fat percentages and body mass index (BMI) and, therefore, could be used as a sensor system for such kind of mechatronic ski binding.

## 2. Materials and Methods

The current study is part of a more than ten-year ongoing study with the aim to understand injury risks in recreational skiing and to examine the potential of wearable sensors as one component of a future safety system. The ultimate goal is the development of a mechatronic ski binding.

### 2.1. Participants

Ethical approval for this study was obtained from the Ethics Commission of the Technical University of Munich (Ref. no. 2022-159-S-NP). Only women over 40 years of age were included in the study because they are considered a group with a high risk of injury in alpine skiing. Moreover, recruiting a homogeneous sample allowed us to increase possible effect sizes by limiting the influence of the variability of individuals. For the entire skiing day, the participants used their own skiing equipment. Characteristics of the participant and the respective ski were measured (body height and weight, body mass index, thigh circumference, local subcutaneous fat percentage of the thighs, ski radius, ski length, ski length to body height ratio) or asked for (age, skiing experience in years) and noted down. Local subcutaneous fat percentage was measured using a skinfold calliper (Harpenden Skinfold Caliper, Baty International, Bowers Group, Bradford, England, UK). According to the manufacturer’s recommendation, a vertical skin fold was used for this purpose on the front of the thigh, between the hip and knee joint with the leg being straight and relaxed.

Moreover, the participants assigned themselves a skier type from 1 to 3, with type 1 corresponding to an unsafe and inexperienced skiing style, type 3 to a very safe and practised skiing style, and type 2 including skiers with an intermediate skiing level which are not classified as skier type 1 or 3. The classification according to ISO 11088:2018 [32] was given for orientation. The self-assigned skier type was corrected, if necessary, by the supervisors (certified ski trainers) based on the observations throughout the day. The preparations of the participants and the equipment took place in a separate and heated room. Body height and body mass were recorded and the BMI was calculated.

### 2.2. Experimental Design

The study was performed in March and April 2022 in the ski area Zugspitze (Garmisch-Partenkirchen, Germany). The day was divided into eight sections preparations, reference runs, free skiing, second reference run, rest during lunch break, third reference run, free skiing, and fourth reference run (Figure 1b. The individual sections were always carried out at the same time of the day and in the same order. Before each skiing task, the participants performed two squats in order to synchronise the EMG and camera data. The first reference run was performed right at the beginning of the skiing day. After two hours of free skiing, the second reference run was performed. Then a 45-minute lunch break followed for regeneration and nutrition. After the lunch break, the third reference run was performed, again followed by two hours of free skiing. Finally, a fourth reference run was performed, thus concluding the skiing day and the measurement. Each day, the data of a maximum of two participants was recorded with each participant being supervised by a separate supervisor. An action camera was installed on one ski to record the skiing. A second camera was carried by the supervisor who continuously filmed the participants while skiing. A GPS watch (Forerunner 920 XT, Garmin, Schaffhausen, Switzerland) was used to track the total distance covered throughout the skiing day (including lift rides).

### 2.3. Skiing Tasks

All four reference runs followed an identical protocol and consisted of four different skiing tasks (Figure 1a): six double plough turns, walking uphill in V steps (including the use of ski poles and arms), six double turns with short radius (about 3 m), six double turns with medium-large radius (about 6 m). The first and the last turn were not included in the data analyses as these turns included the initialisation and finalisation of the skiing task and, therefore, may differ.

### 2.4. EMG Measurement

Thigh muscle activity was recorded using EMG pants (MShort3, Myontec Ltd. Kuopio, Finland) which were available in all sizes. The compressive pants measure the activity of the whole muscle group of the quadriceps as well as the hamstrings using textile electrodes, which have skin contact. According to the manufacturer’s manual, the skin was wetted before putting on the pants and the skin was not specially prepared. The measurement frequency was 1000 Hz throughout the reference runs. For each participant, the correct pant size was chosen according to the manufactures recommendation. After putting on the EMG pants and the participant’s skiing underwear, a five-minute instructed warm-up was performed. This was followed by measuring all four muscle groups’ maximum voluntary contraction (MVC). For recording the MVC, the participants performed an isometric contraction of at least five seconds against a fixed sling (see Figure 2a for the setup to measure the MVC for the quadriceps and Figure 2b for the setup to measure the MVC of the hamstrings). The highest mean of one second of the recording was subsequently used as the norm (MVC-value).

### 2.5. Data Analysis and Statistics

The EMG data were analysed using Matlab (Version R2021b, The MathWorks Inc., Natick, MA, USA). Outlier values in the raw EMG were removed by setting them to zero if they were outside ten times the standard deviation of the complete measurement data. Such singular outlier values were few and could be the result of errors in the Bluetooth data transmission of the raw EMG signal. A 10 Hz high pass filter was used to control for motion artefacts (Matlab function: highpass).

From the raw EMG data set of the reference runs the data of the specific skiing tasks were extracted and a fast Fourier transformation was applied to this extracted data. From the frequency spectrum the mean frequency was calculated thus providing the metric to quantify the shift of mean frequency towards lower values throughout the skiing day. Apart from the analysis in the frequency domain, the extracted data were also processed in the time domain by rectifying and filtering using a moving average with a window size of 50 ms. All data were then normalized using the individual MVC. Further, the ratio of quadriceps and hamstring data (Quad/Ham-ratio) was calculated for each skiing manoeuvre. This was performed separately for the left and right leg.

A two-way ANOVA with repeated measurements of the factors ‘reference run’, ‘skiing task’, and the ‘muscle activity’ (sum over all muscle groups) as the dependent variable was carried out using SPSS (Vers. 27.0, SPSS Inc. Chicago, IL, USA) with a significance level of 0.05. A Greenhouse–Geisser correction of the degrees of freedom was made in case the prerequisite for sphericity was violated (Mauchly test). Moreover, the existence of interaction effects was tested. A post hoc test according to Bonferroni was carried out to subsequently compare all levels of the two factors reference run and skiing task. Since the change in the EMG signal of the participants over the day is decisive in the analysis of muscular fatigue, a mixed ANOVA was used with the factor ‘reference run’ with repeated measurements, the factor ‘skier type’ without repeated measurements, and the dependent variable ‘muscular fatigue’.

## 3. Results

In total, 40 recreational skiers were recruited for this study, of which 38 participants successfully finished all measurements and tasks throughout the day. One participant terminated the study on her own wish at an early stage of the day, so no sufficient data could be collected and analysed. The second one was not able to perform the predefined skiing tasks properly and, therefore, was excluded from the study. The 38 participants (mean age 53.1 ± 5.8 years) included in this study had an average skiing experience of 45 ± 10.2 years with 24 being skier type 2 and 14 being skier type 3. On average, the participants cover a distance of 52.76 ± 4.33 km while skiing and during lift rides over the day. The BMI did not significantly influence the normalized EMG signal measured by the pants (Figure 3). The same was true for the local body fat percentage/local subcutaneous fat percentage (see Appendix A). The correlation between BMI and local subcutaneous fat percentage (Pearson 0.669 *p* < 0.01) and local skinfold thickness (Pearson 0.692 *p* < 0.01) was significant. An overview of the characteristics of the participants is given in Table 1.

### 3.1. Results from the Analysis in the Time-Domain

The muscle activity over all muscle groups was lowest after lunch, in reference run 3 (Figure 4a). The muscle activity of the quadriceps muscles was highest for reference run 1 and continuously decreased in reference run 2 and 3. In reference run 4 it increased again. The muscle activities of the hamstring muscles were comparable for reference run 1 and 2, before decreasing in reference run 3 and again increasing in reference run 4 (an increase of +6 to 17% compared to reference run 3). The only significant difference in muscle activity was found between reference run 1 and 3 for the right quadriceps (*p* = 0.001) but not between the other reference runs. For the other muscle groups, no significant difference was found between the four reference runs.

When comparing the muscle activity between the different skiing tasks, it was lowest for the plough in all muscle groups (Figure 4b). The results for the plough are significant compared to the V steps (*p* < 0.05 for all muscle groups except the left quadriceps), the turns with a small radius (*p* < 0.05 for all muscle groups), and the turns with a medium radius (*p* < 0.05 for all muscle groups except the left hamstrings). The other skiing tasks did not differ significantly from each other in any muscle group.

The quadriceps–hamstring ratio was similar for the left and right leg (mean over all participants). It averaged 4.55 ± 3.15 (normalized to MVC) for the right leg and 4.49 ± 4.76 (normalized to MVC) for the left leg. There was no significant change in the quadriceps to hamstrings ratio throughout the day (Figure 4a). The quadriceps–hamstrings ratio was highest during the skiing task plough (right leg: 5.97 ± 4.16, left leg: 5.60 ± 5.47), followed by the V steps, turns with medium radius, and turns with short radius (Figure 4b). Differences were significant for both legs between the plough and all other skiing tasks and between the turns with medium radii and the turns with short radii.

The muscle activity of the quadriceps of participants of skier type 2 ranged from 111% to 207% of their respective MVC for the right leg and from 85% to 188% for the left leg. For the skier type 3, the quadriceps muscle activity ranged from 71% to 162% of their respective MVC for the right leg and from 79% to 312% for the left leg. The muscle activity of the hamstrings of participants of skier type 2 ranged from 36% to 102% of their respective MVC for the right leg and from 39% to 103% for the left leg. For the skier type 3, the hamstring muscle activity ranged from 19% to 59% of their respective MVC for the right leg and from 24% to 71% for the left leg. Even though the average muscle activities of skier type 3 were smaller than those of skier type 2, this difference was not significant.

The quadriceps–hamstring ratio ranged from 2.87 to 6.32 for skier type 2 and from 3.66 to 8.63 for skier type 3 (difference not significant).

### 3.2. Results from Analysis in the Frequency Domain

A continuous decrease in the median of the EMG frequency from reference run 1 to reference run 3 between 9 and 15% could be observed for all muscle groups (Figure 5 at-top). The frequency then increased again in reference run 4, but remained below the level of reference run 2. The decrease from reference run 1 to 3 was significant in all muscle groups (Figure 5 at-bottom).

The skiing task “plough” significantly showed the highest medial muscle frequency (Figure 5b). This was true for all muscle groups. There were no other significant differences between the skiing tasks.

No interaction effects between the reference runs and the skiing tasks or between the reference runs and the skier type were found in any of the previously mentioned analyses.

Detailed data can be found in the Appendix A.

## 4. Discussion

The aim of this study was to investigate how muscle fatigue and muscle activity in the quadriceps and hamstrings develop during the skiing days of female skiers.

Overall, the muscle activity could be recorded well using textile surface EMG and different skiing tasks (driving style) can be recognized by muscle activity. Quadriceps muscle activity recorded in this study averages 145% (right leg) and 153% (left leg) of the MVC. These results are comparable with the values of 100–180% described in the literature [18,33]. The high values can be explained by the predominantly eccentric muscle work of the knee extensors during alpine skiing.

In this study, no influence of BMI on the normalized EMG signal was observable. This indicates that the EMG pants can be used by a wide range of individuals. This advantageous behaviour of the EMG pants could possibly be a result of the use of large-area electrodes. The BMIs of the participants included in this study were between 18 and 33. As subcutaneous fat dampens EMG signals (even though we found no significant effect for the local subcutaneous fat percentage on the normalized EMG signal in this study), the validity of using these pants should be investigated for even higher BMIs.

Muscle activity and medial frequency decreased between reference run 1 and 3. This frequency decrease in the EMG indicates muscular fatigue [28]. An expected regeneration during the lunch break was not discernible. The observed decrease in muscle activity is rather atypical, as muscle fatigue is associated with an increase in muscle activity [27].

The increase in muscle activity (and medial frequency for the hamstrings) towards the end of the ski day (reference run 4) could be related to changing snow conditions and, therefore, only allows limited interpretation. Since the study was carried out in March and April and the solar radiation was already relatively strong, towards the end of the skiing day the snow was mostly wet and pushed together. This could also have led to a change in skiing technique. Due to the uneven ground and the resulting increased compensatory movements from the legs, the knee extensors and flexors have to apply more force than on a well-groomed slope. Moreover, Kröll et al. [18] observed a change in skiing technique at the end of a three-hour ski day.

Comparing the muscle activity of the two types of skiers, it was lower for type 3 skiers than for type 2 skiers. Assumably, with increasing skiing ability, less muscle power has to be applied with the same skiing style. Only for the left quadriceps, these differences between the skier types were not observed uniformly across all examined variables. Moreover, the differences in muscle activity between the individual forms of skiing are higher in type 3 skiers than in type 2 skiers. That the observed differences are not statistically significant may be due to the small sample sizes of 24 (skier type 2) and 14 (skier type 3) and very large inter-individual variances between the skiers, especially in the left quadriceps muscle group.

A high degree of co-contraction of the hamstring muscles is associated with good stabilization of the knee joint and thus prevention of ACL injuries. Therefore, the quadriceps–hamstring ratio could be an indicator of a well-stabilised knee. Nevertheless, an influence of skiing ability could not be seen in this study which indicates that a better skiing technique does not automatically result in better stabilisation of the knee joint. The skiing task has an impact on the quadriceps–hamstrings ratio, which was highest during the plough and lowest for the turns with a small radius. It can be assumed that the knee is stabilised the worst during the plough and is best stabilized during the short radius turns. The latter may be a result of more active use of the hamstrings in higher dynamic situations (such as in turns with a short radius). The order of the skiing tasks was not randomized and the plough was always the first task to be performed. This limitation may be influential for the plough having the lowest muscle activity level. A consistent change in the quadriceps–hamstrings ratio throughout the day could not be observed. Possible fatigue is therefore not reflected in the quadriceps–hamstrings ratio.

Based on the results of this study, muscle activity measured by EMG pants could be used as an input variable for a mechatronic ski binding algorithm, especially when viewed together with speed. A high level of muscle activity at low speed could indicate an unsafe skiing style or difficult snow conditions. In both cases, it would make sense to adjust the ski binding to lower retention values. Although there is also an increased muscle activity when skiing on even ground or slightly uphill, a smaller retention setting is not to be viewed critically in these cases of skiing on even ground and in good snow conditions, because, in general at low speed, there are no external forces acting on the skier that could lead to the ski binding being released unintentionally.

As discussed above, this study has limitations. Especially a larger number of skiers (also males and a wider age range) should be tested and also the same participants should be measured multiple times to cope with the individual and environmental influences (e.g., snow conditions). If task-specific muscle fatigue is the area of interest, the order of the skiing tasks should be randomized in future studies. We measured MVC only once at the beginning of the skiing day. MVC is known to be subjected to a learning effect. Thus, MVC mean could have been higher in a second or third iteration. Therefore, the fact that we measured the MVC only once is a limitation of our study. However, the effect is minimized as we performed an ANOVA with repeated measurements and performed an intra-individual comparison. A statistical limitation is that a normal distribution according to Kolmogorov–Smirnov or Shapiro–Wilk could not be demonstrated for all variables but ANOVA was nevertheless performed. Since the sample is sufficiently large with *n* = 38, the ANOVA is relatively robust to violations of the normal distribution assumption [34].

## 5. Conclusions

Overall, muscle activity can be recorded well during alpine skiing using textile surface EMG. The BMI (between 18 and 33) did not impact the results significantly allowing the use of such systems by a wide range of individuals. Muscular fatigue can be determined by observing a decrease in the medial frequency. If recorded by a wearable feedback system this could prevent injuries by informing the skier of the higher injury risk. Fatigue is not reflected in the quadriceps–hamstrings ratio.

## Figures and Tables

**Figure 1 ijerph-20-05486-f001:**
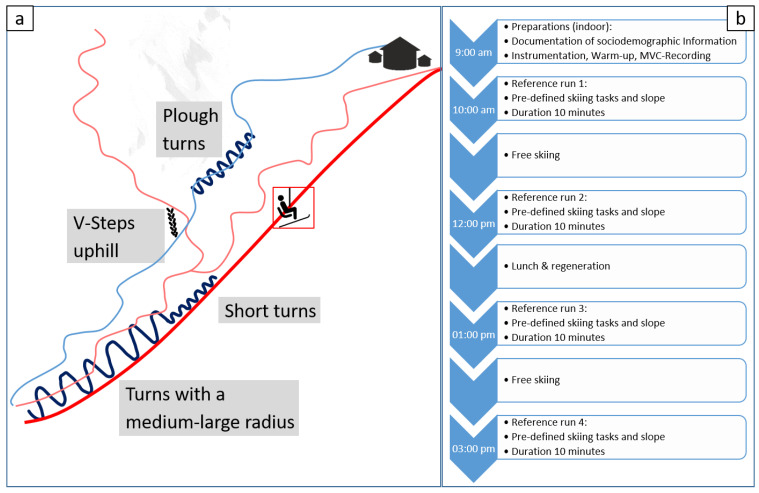
(**a**) Pre-defined skiing tasks and slopes for the reference runs; (**b**) protocol of the measurement day.

**Figure 2 ijerph-20-05486-f002:**
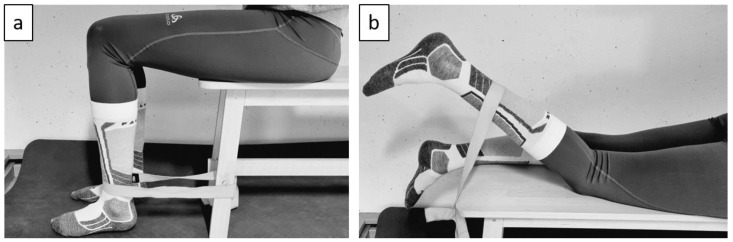
(**a**) Setup to record the MVC of the quadriceps. (**b**) Setup to measure the MVC of the hamstrings.

**Figure 3 ijerph-20-05486-f003:**
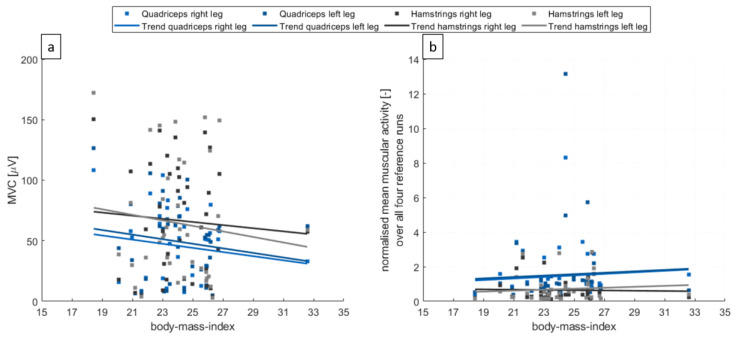
(**a**) The MVC values of the quadriceps and hamstring groups of all participants and the trendline with respect to the participant’s body mass indexes. (**b**) The normalised mean muscular activity over all reference runs of the day for the four muscle groups and all participants.

**Figure 4 ijerph-20-05486-f004:**
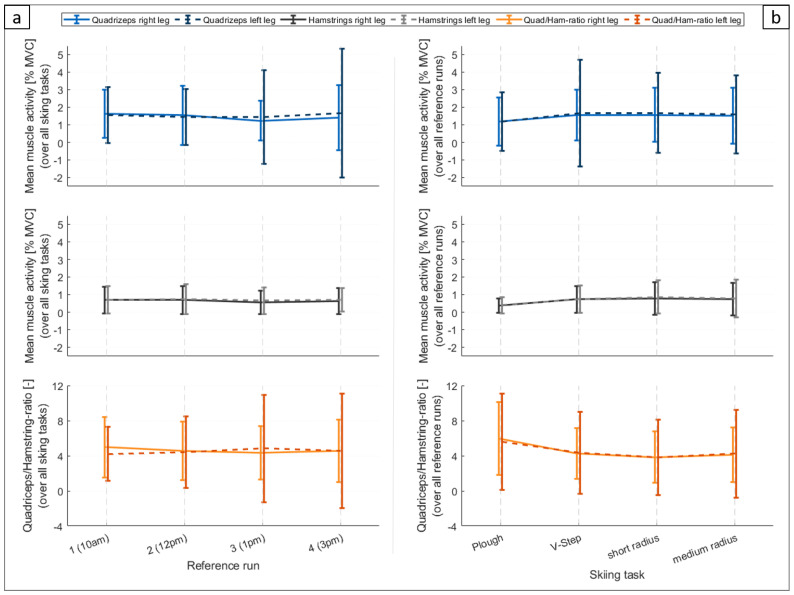
First two lines of graphs: Mean muscle activity in % of the individual reference measurement (maximum voluntary contraction|MVC) for the four different muscle groups. The error bars indicate the standard deviation. Last line of graphs: the ratio of quadriceps/hamstring activation (quad/ham-ratio) is given for the left and right leg. (**a**) mean muscle activity and quad/ham-ratio over all skiing tasks for the four reference runs; (**b**) mean muscle activity and quad/ham-ratio over all four reference runs for the different skiing tasks.

**Figure 5 ijerph-20-05486-f005:**
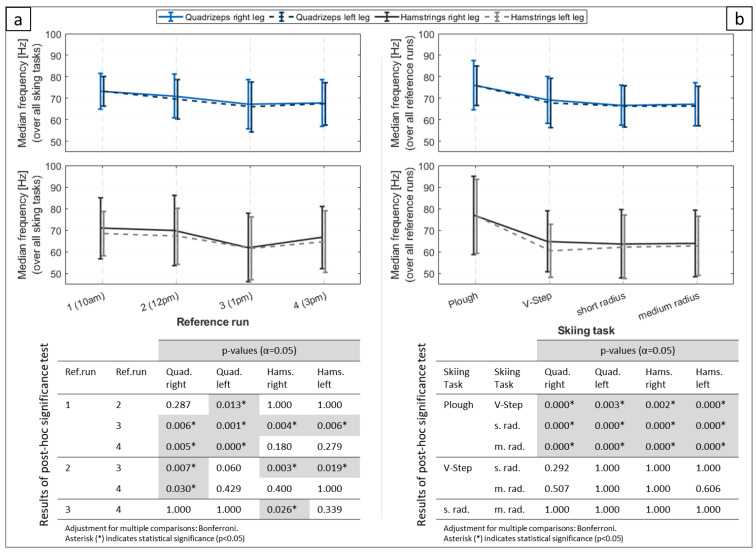
(**a**) graphs: Median frequency of the EMG over all skiing tasks for the four reference runs. The error bars indicate the standard deviation. (**b**) graphs: Median frequency of the EMG over all reference runs for the four different skiing tasks. The error bars indicate the standard deviation. The tables below each pair of graphs give the results of the post hoc significance test with a significance level of α = 0.05.

**Table 1 ijerph-20-05486-t001:** Characteristics of the study participants and their respective ski (*n* = 38).

	min	max	Mean	SD
Age [a]	42	67	53.1	5.8
Body height [m]	1.59	1.86	1.697	0.067
Body weight [kg]	54.8	96.4	69.33	8.98
BMI [kg/m^2^]	18.45	32.59	24.054	2.435
Thigh circumference [cm]	47.0	68.5	56.21	4.15
Body fat percentage [%]	17.2	30.6	23.81	2.79
Ski radius [m]	10.8	17.4	13.47	1.51
Ski length [m]	1.56	1.71	1.641	0.039
Ski length to body height ratio	0.90	1.03	0.970	0.036
Skiing experience [a]	15	63	45.0	10.2
Total distance covered during the day [km]	42.8	65.0	52.76	4.33

## Data Availability

The statistical data presented in this study are available in the Appendix A. Further data are available on request from the corresponding author. The data are not publicly available due to personal rights.

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
