# Peer review of "Muscular Fatigue and Quadriceps-to-Hamstring Ratio in Alpine Skiing in Women over 40 Years"

_ijerph, 2023, doi:10.3390/ijerph20085486_

Round 1

Reviewer 1 Report (Previous Reviewer 2)

The requested changes have been made, paper can be published.

Author Response

Thank you again for your help in improving our manuscript!

Reviewer 2 Report (New Reviewer)

The paper aims to identify muscle activity and muscle fatigue in thigh muscles of middle-aged female skiers over the duration of a full skiing day. A secondary aim is to investigate if EMG measurements using EMG pants are affected by body fat percentage and body mass index (BMI). The main contribution of the paper is the finding that EMG pants can be used to measure valid muscle activity in female thigh muscles regardless of BMI. In addition, using EMG pants, muscle fatigue can be quantified based on changes in median frequency of muscle activity. This enables to use the sensor system as an easily accessible and simple to use tool to measure the level of muscle fatigue in individuals during sports activities, which could help to prevent injuries that are normally attributed to occur due to fatigue. Strengths of the article include the application of EMG pants instead of standard electrode-based EMG measurement systems. The EMG pants are much easier to use and more comfortable for the subjects. The experimental protocol was sound and appropriate for the research objective stated in the introduction. The overall structure of the article is clear and easy to follow.

General and conceptual comments

Introduction:

Regarding the research objective. The gap for the second part of the objective is clear, it will be investigated if EMG pants and thus muscle activity can be used as a sensor system for a mechatronic ski binding. The reasoning or the gap for the first part of the objective is not appropriately prepared.

Generally, some information was missing in the methods and results section. The missing aspects are stated in the following:

Methods:

The method section describes the statistical analysis. It is implicitly stated that the prerequisite for sphericity is tested for, because if the sphericity prerequisite was violated, a Mauchly test was performed. However, no information is provided for the other prerequisites that are needed for an ANOVA with repeated measurements, namely homogeneity of variance and normal distribution within groups. If these prerequisites were also tested for, this should be stated in the methods section. If not, either these assumptions have to be tested or there should be a paragraph included in the discussion which describes the consequences of a violation of these assumptions.

Results:

The skier type is not stated in the characteristics table nor in the rest of the result section. As skier type is a measurement factor in the ANOVA, it should be added in the results section how many subjects were classified into each skier level.

Post-hoc significance test results for mixed ANOVA (measurement factors ‘reference run’ and ‘skier type’) should be added for completion (at least in the supplementary material), as it is included for the two-way ANOVA with measurement factors ‘reference run’, ‘skiing task’ and ‘muscle activity’.

The conclusion states that EMG measurements using EMG pants are not influenced by BMI. This statement is derived from a plot which is included in the supplementary materials. As this statement is one of the key findings of the article, the plot should be included in the main manuscript.

The research objective states that it will be investigated if body fat percentage and BMI affect EMG measurements using EMG pants. In the results, this effect is only investigated for the factor BMI and not the body fat percentage value. As BMI and body fat percentage do not have to correspond, the research question is not fully investigated and answered.

Specific comments

Methods:

·       In line 94 it is stated that the body fat percentage of each subject was measured. However, there is no information about how this value was measured i.e. which method or tool was used. This is relevant in order to be able to evaluate how reliable the value is, since some methods work better than others.

·       Line 94: It is stated that normalized EMG signals measured by EMG pants are not affected by BMI. Some additional argumentation or a further explanation statement would be appropriate, because as stated above, this is declared as one of the main findings of the article, as it is written in the conclusion.

·       Lines 148/149: Figure 2b is written instead of Figure 2a and Figure 2c instead of Figure 2b.

·       Line 150: Why is MVC only measured once? MVC is known to be subjected to a learning effect. Thus, MVC mean could have been higher in a second or third iteration.

Discussion:

·       Line 254: It is not clear from which result the statement can be derived that driving style can be recognized by muscle activity as the result section states that there was no significant difference between average muscle activities of skier type 2 and 3 (stated in line 234).

·       The reasoning in lines 264 – 267 is unclear. It is stated that muscle frequency decreased between reference run 1 and 3 and that this supports finding of other researchers stating that female skiers mainly injure themselves within the first three hours of skiing. But if muscle fatigue increases and also does not decrease after a break at noon, shouldn’t most injuries then occur after the first three hours when fatigue has set in? This seems contradicting.

Figures:

·       In figure 1 the top line of the figure and the figure caption are redundant. They both state the same.

·       Figure 3 has to be adjusted, because it is not easy to read and interpret. The error bars should be symmetrical (so both the positive and the negative value should be shown), because standard deviation is always bi-directional and there is no reason to only show one side (even though it is symmetrical). Same goes for figure 4. In addition, it is not easy to distinguish the different curves within one plot. It is very difficult to distinguish the shades of blue from one another, same goes for the shades of orange.  Also vector graphic format would be appropriate, because the quality of the plot is not good enough (pixelated).

·       A similar problem occurs in Figure 4 and Figure 1 in the supplementary material, the shades of blue are again not distinguishable. Other colours should therefore be chosen.

Author Response

We thank you very much for your review and the time and effort you invested. Your review was exceptionally helpful and well-made. It helped us to improve our work. In the following we address your comments:

General and conceptual comments

Introduction:

Comment: Regarding the research objective. The gap for the second part of the objective is clear, it will be investigated if EMG pants and thus muscle activity can be used as a sensor system for a mechatronic ski binding. The reasoning or the gap for the first part of the objective is not appropriately prepared.

Generally, some information was missing in the methods and results section. The missing aspects are stated in the following:

Answer:
The first gap addresses the effect of fatigue specifically on injury risks in skiing.  This effect is still unclear with contradicting results being published. We added new sentences in the introduction to highlight the benefit of studying fatigue evolution throughout the skiing day.

Methods:

Comment: The method section describes the statistical analysis. It is implicitly stated that the prerequisite for sphericity is tested for, because if the sphericity prerequisite was violated, a Mauchly test was performed. However, no information is provided for the other prerequisites that are needed for an ANOVA with repeated measurements, namely homogeneity of variance and normal distribution within groups. If these prerequisites were also tested for, this should be stated in the methods section. If not, either these assumptions have to be tested or there should be a paragraph included in the discussion which describes the consequences of a violation of these assumptions.

Answer:

Homogeneity of variance is a prerequisite for an ANOVA without repeated measurements. For an ANOVA with repeated measurements, sphericity as a prerequisite is sufficient. It is tested with the Mauchly test and means, that the variances of the differences between factor levels are equal. The test for sphericity, therefore, replaces the test for homogeneity of variance.

References:

Huynh, H. & Feldt, L.S. (1976). Estimation of the Box correction for degrees of freedom from sample data in randomised block and split-plot designs. Journal of Educational Statistics, 1, 69–82

Eid, M., Gollwitzer, M. & Schmitt, M. (2017). Statistik und Forschungsmethoden (5. Aufl.). Beltz.

Although a normal distribution according to Kolmogorov-Smirnov or Shapiro-Wilk could not be demonstrated for all variables, the ANOVA was nevertheless performed. Since the sample is sufficiently large with n = 38, the ANOVA is relatively robust to violations of the normal distribution assumption. We added this information to the limitation section in the discussion.

References:

Blanca, M. J., Alarcón, R., Arnau, J., Bono, R. & Bendayan, R. (2017). Non-normal data: Is ANOVA still a valid option? Psicothema, 29(4), 552–557. https://doi.org/10.7334/psicothema2016.383

Eid, M., Gollwitzer, M. & Schmitt, M. (2017). Statistik und Forschungsmethoden (5. Aufl.). Beltz.

Glass, G. V., Peckham, P. D. & Sanders, J. R. (1972). Consequences of Failure to Meet Assumptions Underlying the Fixed Effects Analyses of Variance and Covariance. Review of Educational Research, 42(3), 237–288. https://doi.org/10.3102/00346543042003237

Harwell, M. R., Rubinstein, E. N., Hayes, W. S. & Olds, C. C. (1992). Summarizing Mon-te Carlo Results in Methodological Research: The One- and Two-Factor Fixed Effects ANOVA Cases. Journal of Educational Statistics, 17(4), 315–339. https://doi.org/10.3102/10769986017004315

Results:

Comment: The skier type is not stated in the characteristics table nor in the rest of the result section. As skier type is a measurement factor in the ANOVA, it should be added in the results section how many subjects were classified into each skier level.

Answer:
You probably missed that. This info was stated right at the beginning of the result section: “The 38 participants (mean age 53.1 ± 5.8 years) included in this study had an average skiing experience of 45 ± 10.2 years and 24 being skier type 2 and 14 being skier type 3.”

Comment: Post-hoc significance test results for mixed ANOVA (measurement factors ‘reference run’ and ‘skier type’) should be added for completion (at least in the supplementary material), as it is included for the two-way ANOVA with measurement factors ‘reference run’, ‘skiing task’ and ‘muscle activity’.

Answer:

As the mixed ANOVA showed no significant interaction effects, no post-hoc test was carried out. We added the information in the result section. We also deleted the information in the methods section that a post-hoc test was performed, as this was incorrect.

Comment: The conclusion states that EMG measurements using EMG pants are not influenced by BMI. This statement is derived from a plot which is included in the supplementary materials. As this statement is one of the key findings of the article, the plot should be included in the main manuscript.

Answer: We moved the plot into the manuscript. We also added an extra plot for the local subcutaneous fat percentage in the supplementary materials.

Comment: The research objective states that it will be investigated if body fat percentage and BMI affect EMG measurements using EMG pants. In the results, this effect is only investigated for the factor BMI and not the body fat percentage value. As BMI and body fat percentage do not have to correspond, the research question is not fully investigated and answered.

Answer:

The BMI was used as a proxy measure for body fat percentage. The correlation between BMI and local body fat percentage/local subcutaneous fat percentage (Pearson 0.669 p< .01) and local skinfold thickness (Pearson 0.692 p< .01) was significant. We added the information to the results section.

Specific comments

Methods:

Comment: ·       In line 94 it is stated that the body fat percentage of each subject was measured. However, there is no information about how this value was measured i.e. which method or tool was used. This is relevant in order to be able to evaluate how reliable the value is, since some methods work better than others.
Line 94: It is stated that normalized EMG signals measured by EMG pants are not affected by BMI. Some additional argumentation or a further explanation statement would be appropriate because as stated above, this is declared as one of the main findings of the article, as it is written in the conclusion.

Answer:
To be more concise, we substituted “body fat percentage” with “local subcutaneous fat percentage of the thigh”. We added some information in the methods section. The local body fat percentage was determined by measuring the local skinfold thickness on the inside of the thighs of both legs in accordance with the manufacturer’s recommendation. The mean of both measured values was calculated and compared with a scale provided by the manufacturer of the Skinfold Caliper (depending on the age of the subject) to determine the local body fat percentage.

We now elaborate more on the subject of “BMI influence and EMG pants” in the discussion section:

“In this study, no influence of BMI on the normalized EMG signal was observable. This indicates that the EMG pants can be used by a wide range of individuals. This advantageous behavior of the EMG pants could possibly be a result from the use of large-area electrodes. The BMI of the participants included in this study was between 18 and 33. As subcutaneous fat dampens EMG signals (even though we found no significant effect for the local subcutaneous fat percentage on the normalized EMG signal in this study), the validity of using these pants should be investigated for even higher BMIs.”

Comment: ·       Lines 148/149: Figure 2b is written instead of Figure 2a and Figure 2c instead of Figure 2b.

Answer:
Thank you for reading this so thoroughly. We corrected this

Comment: ·       Line 150: Why is MVC only measured once? MVC is known to be subjected to a learning effect. Thus, MVC mean could have been higher in a second or third iteration.

Answer:
The fact that we measured der MVC only once is a limitation of our study. However, the effect is minimized as we performed an ANOVA with repeated measurements and did intraindividual comparison. We added this to the limitation section in the discussion.

Discussion:

Comment: ·       Line 254: It is not clear from which result the statement can be derived that driving style can be recognized by muscle activity as the result section states that there was no significant difference between average muscle activities of skier type 2 and 3 (stated in line 234).

Answer:
Our statement was not clearly formulated. The sentence refers to the skiing task. We rephrased the sentence to:

“Overall, the muscle activity could be recorded well using textile surface EMG and different skiing tasks (driving style) can be recognized by muscle activity.”

Comment: ·       The reasoning in lines 264 – 267 is unclear. It is stated that muscle frequency decreased between reference run 1 and 3 and that this supports finding of other researchers stating that female skiers mainly injure themselves within the first three hours of skiing. But if muscle fatigue increases and also does not decrease after a break at noon, shouldn’t most injuries then occur after the first three hours when fatigue has set in? This seems contradicting.

Answer:

Thank you for this comment. We again checked the literature referenced (Ruedl et al. 2010) in this section and agree that the reasoning is not sound. As we also were not correctly stating the findings of the reference, we deleted the sentence.

Side note:

By reading the manuscript of Ruedl et al, the following considerations may be of interest:

RUEDL and co-authors have chosen “Are ACL Injuries Related to Perceived Fatigue in Female Skiers?” as the title of their 2010 publication in ASTM Int. Vol7, Issue3.
Consequently, they have asked the ACL-injured female skiers and the controls, which level of fatigue they have perceived.

It is interesting to note, that only 27% of the control group did not feel fatigued (compared to 81% of the injured). This logically means that 73% of the controls felt fatigued when being asked.

This rises two questions:

  1. Is the instant of time when the subjects were asked about their perceived fatigue, comparable? Whereas the ACL injured were asked retrospectively in the hospital within two days after the accident, the controls were asked on the slope after lunch time and at the late afternoon. This means for instance, that the percentage of controls having been skiing for more than two hours (and even more if they had started their skiing day at 8 or 9am) is higher than for the patient group.
  2. As the controls’ interview times were not matched to those of the injured (the control interviews took place on five different days over a period of 2 months), it is possible, that the slope conditions have been different. So, if -for instance- a skier in the control group was interviewed at the end of this 2 months period and in the late afternoon, the snow might have been slushy and worse groomed, and thus more exhaustion/fatigue may have been perceived.

Remains the finding, that the mean time of skiing in patients and controls was statistically different (Fig.1 in Ruedl et al., 2010). But what would we expect, if we ask a group of people, who had an accident and thus “finished” skiing in comparison to a control group, which was free to choose when to stop (and being asked their skiing duration in the late afternoon)?

With the aforementioned concerns regarding methodology (Ruedl et al. have mentioned it in their discussion) and the results in mind, that patients were significantly smaller, heavier, and had a higher BMI than the controls, we think that the conclusion the authors have made in their abstract, stating “…fatigue seems no major risk factor for an ACL injury in female recreational skiing” is not supported by the study. At least they should have added the word “perceived” or better formulated their conclusion with lot more caution.

And this brings us to a major question and the added knowledge from our investigation: It may be the case that skiers do not perceive fatigue (at least upcoming fatigue)? However, the objective metric (a drop of EMG’s mean frequency) is able to detect it and could, therefore, feedback fatigue to the skier.

Figures:

Comment: ·       In figure 1 the top line of the figure and the figure caption are redundant. They both state the same.

Answer:
We deleted the top line of the figure to eliminate the redundancy.

Comment: ·       Figure 3 has to be adjusted, because it is not easy to read and interpret. The error bars should be symmetrical (so both the positive and the negative value should be shown), because standard deviation is always bi-directional and there is no reason to only show one side (even though it is symmetrical). Same goes for figure 4. In addition, it is not easy to distinguish the different curves within one plot. It is very difficult to distinguish the shades of blue from one another, same goes for the shades of orange.  Also vector graphic format would be appropriate, because the quality of the plot is not good enough (pixelated).

A similar problem occurs in Figure 4 and Figure 1 in the supplementary material, the shades of blue are again not distinguishable. Other colours should therefore be chosen.

Answer:
Both figures were adjusted and redesigned according to your recommendations. We increased the color contrast. When uploaded by us, the figures have a high quality. Most are even in vector format. We will clarify this with the editorial office and will provide figures in quality according to the journal’s needs.

Reviewer 3 Report (New Reviewer)

Greetings and thank you for choosing me as a reviewer of the article entitled “Muscular Fatigue and Quadriceps-to-Hamstrings Ratio in 2 Alpine Skiing in Women Over 40 Years”. I would like to inform you that the title of the article is very interesting and can be used as a reference for future readers and researchers. The following are suggested to improve the quality of the article:

1-      In the summary part of the article, several sentences are written about the research field, but there is no reference to the samples, measurement methods and variables considered to investigate the effect of the treatment. Please rewrite the summary section taking into account the omitted items.

2-      In lines 51-51, you mentioned the existence of contradiction by referring to three references, please explain the existing contradiction a little. “The effect of fatigue specifically on injury risk in skiing, however, is still unclear with contradicting results being published [23–26].”

3-      In lines 75 to 79, the researcher mentioned one of his goals to test pants equipped with electromyography electrodes. Is it the only test in your mind? Have you not compared with another device or method during this research?

4-      In the materials and methods section (lines 80-103), please indicate whether the samples had a history of falling resulting in injury during skiing or not?

5-      In lines 141-142, you mentioned that before wearing the pants equipped with electrodes, the skin was wetted. Wouldn't this moisture be lost and dried during a day of skiing?

6-      In line 142, the spelling of the word "before" should be corrected.

7-      In lines 180-188, please indicate the reason why two of the samples left your research plan.

Author Response

We thank you very much for your review and the time and effort you invested. It helped us to improve our work. In the following we address your comments:

Comment 1: In the summary part of the article, several sentences are written about the research field, but there is no reference to the samples, measurement methods and variables considered to investigate the effect of the treatment. Please rewrite the summary section taking into account the omitted items.

Answer:

If with the summary you refer to the abstract we mentioned the sample size, measurement methods, and investigated variables. A more detailed description is not possible as we already changed the abstract a few times and are restricted to the word limit given by the journal.
Further, this information is mentioned again in the discussion. So if you are referring to the conclusion, please let us know if it’s not sufficient to mention those points in the abstract and discussion.

Comment 2: In lines 51-51, you mentioned the existence of contradiction by referring to three references, please explain the existing contradiction a little. “The effect of fatigue specifically on injury risk in skiing, however, is still unclear with contradicting results being published [23–26].”

Answer:

We rewrote this paragraph in more detail.

The effect of fatigue specifically on injury risk in skiing, however, is still unclear. Several authors report different relations between fatigue and injuries, either finding no relation between those factors or that fatigue is a risk factor for injuries. [23–26]. Many of these studies show limitations in the methodological approach, e.g., the use of retrospective subjective questionnaires to ask the injured skiers in the hospital about their perceived fatigue within several days after the incident [24,25]. A limitation of such questionnairs is that it may be possible that skiers do not perceive fatigue (at least upcoming fatigue) due to an excited state. It is also unclear, how fatigue evolves throughout the skiing day and if regenerative breaks, like the lunch break, have a positive effect. This knowledge could help to create awareness of such facts for example in ski school lessons or by means of wearable feedback systems. A wearable system providing an objective metric of fatigue would also allow investigation of the relationship between fatigue and injury risk without the aforementioned limitation.”

Comment 3: In lines 75 to 79, the researcher mentioned one of his goals to test pants equipped with electromyography electrodes. Is it the only test in your mind? Have you not compared with another device or method during this research?

Answer:
As the pants are already validated and tested by previous researchers our main goal was to focus on different levels of local subcutaneous fat percentage (which had a high correlation with the BMI), considering the influence of a higher fat percentage on the EMG signal amplitude.

Validation studies:
Finni, T., Hu, M., Kettunen, P., Vilavuo, T. & Cheng, S. (2007). Measurement of EMG activity with textile electrodes embedded into clothing. Physiological measurement, 28(11), 1405–1419. https://doi.org/10.1088/0967-3334/28/11/007

Bengs, D., Jeglinsky, I., Surakka, J., Hellsten, T., Ring, J. & Kettunen, J. (2017). Reliability of Measuring Lower-Limb-Muscle Electromyography Activity Ratio in Activities of Daily Living With Electrodes Embedded in the Clothing. Journal of sport rehabilitation, 26(4). https://doi.org/10.1123/jsr.2017-0019

Colyer, S. L. & McGuigan, P. M. (2018). Textile Electrodes Embedded in Clothing: A Practical Alternative to Traditional Surface Electromyography when Assessing Muscle Excitation during Functional Movements. Journal of Sports Science & Medicine, 17(1), 101–109.

Comment 4: In the materials and methods section (lines 80-103), please indicate whether the samples had a history of falling resulting in injury during skiing or not?

Answer:

No such data was investigated or collected. The inclusion criteria for the participants were to have no current injuries and be in good health condition.

Comment 5: In lines 141-142, you mentioned that before wearing the pants equipped with electrodes, the skin was wetted. Wouldn't this moisture be lost and dried during a day of skiing?

Answer:

The pants were moistened according to the manufacturer’s (Mynotec Ltd., Kuopio FIN) recommendation at the beginning of the measurements. During activities like skiing the body tends to sweat (especially in the warmer conditions at the time the study was conducted, march-april). This will keep up with the moistening in the morning.

Comment 6: In line 142, the spelling of the word "before" should be corrected.

Answer:

Thank you. We corrected this.

Comment 7: In lines 180-188, please indicate the reason why two of the samples left your research plan.

Answer:

One participant terminated the study on her own wish at an early stage of the day, so no sufficient data could be collected and analyzed.

The second one wasn’t able to perform the predefined skiing tasks properly and therefore the measured data were extracted. We added this information to the result section.

Round 2

Reviewer 2 Report (New Reviewer)

Brief summary

The paper aims to identify muscle activity and muscle fatigue in thigh muscles of middle-aged female skiers over the duration of a full skiing day. A secondary aim is to investigate if EMG measurements using EMG pants are affected by body fat percentage and body-mass-index (BMI). The main contribution of the paper is the finding that EMG pants can be used to measure valid muscle activity in female thigh muscles regardless of BMI. In addition, using EMG pants, muscle fatigue can be quantified based on changes in median frequency of muscle activity. This makes it possible to use the sensor system as an easily accessible and simple to use tool to measure the level of muscle fatigue in individuals during sports activities which could help to prevent injuries that are normally attributed to occur because of fatigue. Strengths of the article include the application of EMG pants instead of standard electrode-based EMG measurement systems. The EMG pants are much easier to use and more comfortable for the subjects. The experimental protocol was sound and appropriate for the research objective stated in the introduction. The overall structure of the article is clear and easy to follow.

General concept comments

Overall, all mentioned comments have been dealt with and were answered appropriately. I agree with all changes that were made to the manuscript.

The research gap is now much more clearly elaborated and defined. All missing information in the methods (i.e. body fat percentage measurement tool) and result section (i.e. figure 3 was moved from supplementary material to text) was added. The correlation between BMI and local subcutaneous fat percentage was added which explains why BMI can be used as a proxy measure for body fat percentage. Missing limitations in the discussion section were also added.

The new design of figure 4 improves the readability and makes it easier to interpret now that the information was split into 6 subplots overall. It is also easier to distinguish the new colours, as there is a better contrast now.

Regarding the following comment:

Homogeneity of variance is a prerequisite for an ANOVA without repeated measurements. For an ANOVA with repeated measurements, sphericity as a prerequisite is sufficient. It is tested with the Mauchly test and means, that the variances of the differences between factor levels are equal. The test for sphericity, therefore, replaces the test for homogeneity of variance.

Thank you for providing references in regard to this topic. Based on these, I agree that sphericity is sufficient as a prerequisite.

I understand that the ANOVA is robust to this violation with your sufficiently large sample size. Nevertheless, I appreciate that you added a paragraph with this information in the discussion section, as I think it is important to report the violation for the sake of completeness.

Additional specific comments

Regarding figure 4:

·       In the provided version of the revised manuscript, it seems like the darker shade of orange on the lower left side of the plot depicts the old colour (of the previous version) as the shades of oranges in the lower right side of the figure have a much higher contrast in comparison.

·       The error bars in the upper (blue) subplots are outside the y-axis limit.

Author Response

Thank you again for reviewing our manuscript and the valuable feedback.
We corrected figure 4. Thank you!

Reviewer 3 Report (New Reviewer)

Changes are acceptable.

Author Response

Thank you again for reviewing our manuscript!

This manuscript is a resubmission of an earlier submission. The following is a list of the peer review reports and author responses from that submission.

Round 1

Reviewer 1 Report

Dear authors, the manuscript is especially interesting for the task-oriented evaluations seen in figure 1, but methodologically and from an expository point of view, the manuscript needs a major and in-depth revision before being able to evaluate its results and discuss it.

In the introduction put a background of study. Take away the possibility of using electromyography, but why use it. In the goal, avoid abbreviations, among other things for only 2 words. I would suggest describing the high risk of knee injury in senior athletes, especially ACL injury in women compared to men, and that the relative risk is related to phenomena such as muscle fatigue and knee stabilizer muscle activity. “Therefore, this study aimed to…”

13 remove 38 is a result, put what kind of subjects you wanted to include.. professionals? retired athletes? with at least 10 years of experience, but aged over 40?

14, albeit synthetic, please state which technical gestures were requested.

15 superfluous BMI, EMG (among other things surface?) of which muscle?

16 methodology no results

17 methodology on how to analyze MF, what physical data were reported, during which task mentioned in the methods? Data is missing

19 in the methods you don't talk about lunch breaks, why have you studied it?

20 in general it is true that (surface?) EMG is useful as an investigation in the study of risk, but in this manuscript why?

21-22 conclusion that does not recall the results of the study

23 muscle fatigue

To remove, describe the background regarding the role of these muscles

50 but you are using surface EMG! missing references..

53 not appropriate, for a scientific journal. That is a future intervention of the manuscript to evaluate muscle activity, it is not appropriate to call it a safety equipment like the helmet.

54-56 this is the real background, but references in the literature are totally missing has anyone else studied this approach yet? what kind of muscles? with limitation? why did you set the next goal? why did you experiment and conduct this study?

60 senior athletes?

65 approval at the end of the paragraph. Study design first, timepoints. For participants a separate paragraph exclusively with the eligibility of the sample..

80 results

the methods must be structured as design, population, intervention (evaluation procedures), outcome (objective measurements used) and statistical analysis

Surface EMG.. which muscles were evaluated, there is no bibliographic reference… which reference did you use for the Q:H ratio?

Author Response

Dear Reviewer, we thank you for the time and effort invested in this manuscript. It helped to improve the work. In the following you find the answers to your comments and questions and a description of the measures taken. Thank you again.

Dear authors, the manuscript is especially interesting for the task-oriented evaluations seen in figure 1, but methodologically and from an expository point of view, the manuscript needs a major and in-depth revision before being able to evaluate its results and discuss it.

In the introduction put a background of study.

Q: Take away the possibility of using electromyography, but why use it. In the goal, avoid abbreviations, among other things for only 2 words. I would suggest describing the high risk of knee injury in senior athletes, especially ACL injury in women compared to men, and that the relative risk is related to phenomena such as muscle fatigue and knee stabilizer muscle activity. “Therefore, this study aimed to…”

A: We assume that you are referring to the abstract of the manuscript. We reworded the abstract according to your recommendations. The abbreviations were necessary to keep the word count limit for the abstract, which is really tight.

Q: 13 remove 38 is a result, put what kind of subjects you wanted to include.. professionals? retired athletes? with at least 10 years of experience, but aged over 40? 14, albeit synthetic, please state which technical gestures were requested. 15 superfluous BMI, EMG (among other things surface?) of which muscle?

16 methodology no results

17 methodology on how to analyze MF, what physical data were reported, during which task mentioned in the methods? Data is missing

19 in the methods you don't talk about lunch breaks, why have you studied it?

20 in general it is true that (surface?) EMG is useful as an investigation in the study of risk, but in this manuscript why?

21-22 conclusion that does not recall the results of the study

23 muscle fatigue

A: We tried to implement all your comments referring to the abstract and now have a new abstract. Unfortunately, the word count limit of 200 words did not allow us to include all details. All your comments have been addressed in the main sections of the manuscript.

To remove, describe the background regarding the role of these muscles

Q: 50 but you are using surface EMG! missing references..

A: We included two more references with respect to the validation and use of such EMG pants

  1. Finni, T.; Hu, M.; Kettunen, P.; Vilavuo, T.; Cheng, S. Measurement of EMG activity with textile electrodes embedded into clothing. Physiol. Meas. 2007, 28, 1405–1419, doi:10.1088/0967-3334/28/11/007.
  2. Colyer, S.L.; McGuigan, P.M. Textile Electrodes Embedded in Clothing: A Practical Alternative to Traditional Sur-face Electromyography when Assessing Muscle Excitation during Functional Movements. J. Sports Sci. Med. 2018, 17, 101–109.

Q: 53 not appropriate, for a scientific journal. That is a future intervention of the manuscript to evaluate muscle activity, it is not appropriate to call it a safety equipment like the helmet.

A: We removed the scentence.

Q: 54-56 this is the real background, but references in the literature are totally missing has anyone else studied this approach yet? what kind of muscles? with limitation? why did you set the next goal? why did you experiment and conduct this study?

A: To our knowledge, there is no other research published with respect to using surface EMG data as input for an injury prevention or feedback system. This was the ultimate motivation of this study. As described in the manuscript, the influence of muscular fatigue and muscle activity of the thighs has been studied. References are now given in the manuscript.

Q: 60 senior athletes?

A: Not particular. As described in the manuscript, we focused on women over 40 years of age for two reasons: 1) this group is known to be at higher risk of injury; 2) recruiting a homogeneous sample allowed us to increase possible effect sizes by limiting the influence of the variability of individuals

Q: 65 approval at the end of the paragraph. Study design first, timepoints. For participants a separate paragraph exclusively with the eligibility of the sample… the methods must be structured as design, population, intervention (evaluation procedures), outcome (objective measurements used) and statistical analysis

A: Thank you for this comment. We recorded the methodology section according to your comment and find it much improved and easier to read.

Q: Surface EMG.. which muscles were evaluated, there is no bibliographic reference… which reference did you use for the Q:H ratio?

A: References were added. See comment above.

Reviewer 2 Report

It is not clear why the authors measured local subcutaneous fat percentage and thigh circumference.

The influence of BMI on the EMG signal was observed in this study, subcutaneous fat or circumference is not mentioned anywhere in the results and discussion.

In such a situation, data on subcutaneous fat and circumference in the methods (or elsewhere) are redundant.

Figure 4, the table in the lower left corner (hams. left, las ref run) has data 0.339*

If the authors are referring to p values, the * symbol is redundant

This paper has two important goals: the prevention of skiing injuries in middle-aged women and the usability of new technology (EMG pants) for such research.

Perhaps the metric characteristics of the new technology should have been published before, and then the possibilities of EMG pants in diagnostics. This way the paper is a little confused.

The conclusion in the summary emphasizes the possibilities of new technology, and the conclusion in the paper emphasizes injury prevention. That needs to be coordinated.

It is mentioned in several places sensor system for a mechatronic ski binding.

This possibility was not the subject of measurement, perhaps it is enough to mention once it in the introduction.

Author Response

Dear Reviewer, we thank you for the time and effort invested in this manuscript. It helped to improve the work. In the following you find the answers to your comments and questions and a description of the measures taken. Thank you again.

Q: It is not clear why the authors measured local subcutaneous fat percentage and thigh circumference. The influence of BMI on the EMG signal was observed in this study, subcutaneous fat or circumference is not mentioned anywhere in the results and discussion. In such a situation, data on subcutaneous fat and circumference in the methods (or elsewhere) are redundant.

A: Thank you for this comment. We removed Figure 1a and all references on the subcutaneous fat and thigh circumference in the text. We recorded the subcutaneous fat and thigh circumference to test their influence of them on the EMG signal. The idea was, that the BMI, may not be a good indicator of the local subcutaneous fat at the thighs. There was no significant influence and the results of the local subcutaneous fat were comparable to the results of the BMI.

Q: Figure 4, the table in the lower left corner (hams. left, las ref run) has data 0.339*.If the authors are referring to p values, the * symbol is redundant.

A: Thank you for reading the manuscript thoroughly. We corrected the mistake.

Q: This paper has two important goals: the prevention of skiing injuries in middle-aged women and the usability of new technology (EMG pants) for such research. Perhaps the metric characteristics of the new technology should have been published before, and then the possibilities of EMG pants in diagnostics. This way the paper is a little confused.

A: EMG Pants are not new and have been used in other disciplines before, but not in skiing (to our knowledge). We included two more references with respect to the validation and use of such pants

  1. Finni, T.; Hu, M.; Kettunen, P.; Vilavuo, T.; Cheng, S. Measurement of EMG activity with textile electrodes embedded into clothing. Physiol. Meas. 2007, 28, 1405–1419, doi:10.1088/0967-3334/28/11/007.
  2. Colyer, S.L.; McGuigan, P.M. Textile Electrodes Embedded in Clothing: A Practical Alternative to Traditional Sur-face Electromyography when Assessing Muscle Excitation during Functional Movements. J. Sports Sci. Med. 2018, 17, 101–109.

Q: The conclusion in the summary emphasizes the possibilities of new technology, and the conclusion in the paper emphasizes injury prevention. That needs to be coordinated.

A: We coordinated both conclusions.

Q: It is mentioned in several places sensor system for a mechatronic ski binding. This possibility was not the subject of measurement, perhaps it is enough to mention once it in the introduction.

A: It was not the main goal of this study, but as mentioned in the introduction and the aim of this study, the motivation of this work is to see, if EMG and particularly EMG pants can be used in a mechatronic system. Therefore, we would like to keep the references to the mechatronic system as this emphasizes the practical relevance of this work.

Round 2

Reviewer 1 Report

Remove 13 and 14.

18 needless to recall BMI, how did you measure fatigue?

19 is a repetition of the method, remove. Describe some result with data! (among other things, also in the manuscript, there is some p value, but there are no comparisons with SD, IQR.. effect size..)

24 fatique?

Regarding the methods, as I had already recommended: put subparagraphs Design, Population, Intervention, Outcome and finally Statistical Analysis…

There is a sketch of the design, then immediately a description of the intervention with the path, then the population, again the EMG. Path and sensorized shorts are an intervention.

190 “The mean frequency was calculated to test for muscular fatigue development throughout the skiing day.” Has anyone already proposed a similar approach in the literature? Fatigue is a primary outcome, why didn't you use a validated rating scale? If you are the first to propose it, you should evaluate intra and inter rater reliability.. provide model description for reproducibility…

If not, remove the word "fatigue from the title"..

In the methods do not make the reader understand what "skier type 2" is, this definition is not obvious..

321 with fewer than 20 participants it would be more appropriate to use non-parametric tests

The standard deviations are missing from the figures

398 by convention the study objective is paraphrased to start the discussion, then the results of your study are described.. Actually “Overall, the muscle activity parameter during alpine skiing could be recorded well using textile surface EMG and differences in driving style can be recognized by muscle activity”, unfortunately it is not even so true, because you have not provided proof of reliability..

410 Literature reference of indirect measurement of fatigue?

“The observed decrease in muscle activity is 413 rather atypical.” Bibliographic reference on the typical phenomenon..

445 there is a serious lack of bibliographic references for solid statements.. every phenomenon seems to be taken for granted

Even if in the discussion you slavishly pointed out the limitations of the study, unfortunately by convention a limitation subparagraph cannot be missing at the end of the discussion
